# Fabricating High Aspect Ratio Amorphous Alloys Microgrooves by Using Periodically Thinning Jet Electrochemical Milling Method

**DOI:** 10.3390/mi16090979

**Published:** 2025-08-26

**Authors:** Yahui Li, Pingmei Ming, Dongdong Li, Rongbo Zhao, Shen Niu

**Affiliations:** School of Mechanical and Power Engineering, Henan Polytechnic University, Jiaozuo 454003, China; 19939962375@163.com (Y.L.); li863964931@163.com (D.L.); rongbozone@163.com (R.Z.); ns2019@hpu.edu.cn (S.N.)

**Keywords:** jet electrochemical milling, periodically thinning jet size, electrochemical machining, high aspect ratio, amorphous alloys

## Abstract

Jet electrochemical milling (JECM) offers significant advantages for fabricating fine grooves and slits in thin-walled, low-rigidity, and heat-sensitive metallic materials, such as amorphous alloys, owing to its operational flexibility, lack of material constraints, and superior surface quality. Nevertheless, conventional JECM techniques for groove machining encounter limitations including excessive overcut, restricted ability to produce microstructures with high depth-to-width ratios, and reduced machining accuracy. To address these issues, this study proposes an innovative approach termed the periodically thinning jet electrochemical milling (PT-JECM) method. This method involves initially generating a shallow microgroove through a single pass using the original nozzle diameter, followed by successive milling passes with progressively smaller nozzle diameters based on the preformed groove. Comparative analysis with traditional JECM methods reveals that this strategy significantly improves the etching factor from 1.896 to 4.318, corresponding to a 128% enhancement. Furthermore, it markedly decreases the slot width increase from 275 μm to 1 μm and improves the aspect ratio from 0.51 to 0.83, representing a 63% increase, enabling the precision machining of large aspect ratio holes and slot structures.

## 1. Introduction

Metallic glass (MG), also known as amorphous alloy, is characterized by a disordered atomic arrangement that minimizes the presence of microstructural defects such as grain boundaries, vacancies, and dislocations. As a result, metallic glasses (MGs) generally exhibit superior mechanical properties relative to conventional crystalline metals, including enhanced hardness, strength, toughness, and improved resistance to wear and corrosion [1]. These attributes render MGs particularly suitable for applications in aerospace and medical device industries, where materials are often subjected to demanding operational conditions [2]. Nonetheless, MGs typically display brittleness and hardness at low temperatures, while becoming soft and ductile at intermediate to elevated temperatures near their glass transition temperature. This thermal behavior poses challenges for processing MGs using traditional mechanical and thermal machining techniques. In this context, thermoplastic forming has emerged as an effective manufacturing approach for fabricating MG structures and components, leveraging the inherent softening and deformability of MGs upon heating [3]. For example, Inoue et al. [4] successfully produced a palladium-based MG gear with a diameter of 4 mm and a modulus of 0.3, illustrating the material’s excellent glass-forming ability and extensive undercooled liquid region. Similarly, Ma et al. [5] fabricated platinum-based MG microchannel structures with dimensions of 19 μm in depth and 18 μm in width. Despite these advances, thermoplastic forming techniques encounter limitations in fabricating complex geometries, such as pocket-shaped cavities and intricate topographical features, thereby constraining their capacity to fulfill the diverse engineering requirements of metallic glasses. Moreover, the precise temperature control necessary during thermoplastic forming further restricts the widespread adoption of this manufacturing method [6].

Consequently, researchers have investigated both mechanical machining techniques and nontraditional machining methods for the fabrication of MGs. Bakkal et al. [7,8,9] demonstrated that conventional mechanical cutting of zirconium-based MGs yields lower surface roughness relative to alternative approaches. Their findings indicated that, under identical cutting conditions, the cutting forces exerted on metallic glasses were comparable to those observed in aluminum alloys and stainless steels; however, high-speed cutting of MGs was found to induce crystallization. Hsieh et al. [10] reported the formation of crystalline ZrC and TiC phases within the recast layer during electro-discharge machining (EDM) of zirconium-based MGs. Yan et al. [11,12] examined the crystallization behavior of zirconium-based MGs subjected to EDM and observed that crystallization was confined to the surface layer when very low discharge energy was applied. Lu et al. [13] investigated the application of femtosecond laser processing on MGs, revealing that non-crystallized micropores and microgrooves could be fabricated under optimally controlled parameters.

In contrast to the aforementioned subtractive methods, electrochemical machining (ECM) operates on the principle of electrochemical dissolution rather than mechanical removal or thermal effects, thereby offering inherent advantages for the fabrication of heat-sensitive materials such as MGs. ECM is characterized as a low-temperature manufacturing process, typically conducted below 40 °C, which significantly mitigates the risk of crystallization in machined MGs [14,15]. Furthermore, as a non-contact process, ECM does not induce residual stresses, recast layers, or heat-affected zones, and its machinability is independent of the mechanical properties of the workpiece material, rendering it particularly suitable for hard-to-cut materials. To date, several ECM techniques have been developed for processing MGs, including electrochemical milling [16], electrochemical wire cutting, and jet electrochemical machining. Electrochemical milling employs a rod-shaped or ball-shaped cathode as the tool, with machining operations analogous to CNC milling [17]. Liu et al. [18] fabricated complex microgroove structures via electrochemical milling by implementing a strategy that initiates with axial cutting depth followed by lateral milling. Zeng et al. [19] utilized a high-frequency ultra-narrow pulse microelectrochemical wire-cutting process, wherein a reciprocating or unidirectionally moving fine metal wire serves as the tool cathode [20], to produce complex microstructures on amorphous alloys. Jet electrochemical machining (JECM), which utilizes a high-velocity electrolyte jet ejected from an anodic nozzle as the machining tool, has been demonstrated as a highly flexible microfabrication technique for producing microholes, microdimples, microgrooves, and microchannels [21]. As a nontraditional micromilling or microdrilling method, JECM exhibits several comparative advantages in the microfabrication of metallic glasses and other thermally sensitive metallic materials [22,23].

Jet electrochemical machining (JECM) generally operates via two primary modes: drilling and milling [24]. In JECM drilling, a continuous electrolyte jet is directed at a fixed location on the workpiece, resulting in the formation of dimples or holes. This technique is typically employed for the sequential production of arrays of holes or dimples. Conversely, JECM milling utilizes an anodic dissolution mechanism to remove material, offering a potentially cost-effective solution for fabricating grooves, narrow slits, and cavities in hard-to-machine materials. However, the process is limited by relatively low machining accuracy and a restricted maximum achievable aspect ratio (depth-to-width ratio), which constrain its broader applicability.

To address these limitations, several unconventional strategies have been proposed. For instance, methods involving the encapsulation of the electrolyte jet with a gas film [25] or an electrically insulating liquid [26], as well as techniques employing internal jetting combined with external suction of the electrolyte [27], have been developed. Chen et al. [25] utilized a porous mask cathode to optimize electrolyte flow and current distribution, thereby enhancing the shaping accuracy of microgroove bottoms and achieving an aspect ratio of 0.14. Zhai et al. [28] introduced a coaxially assisted ultrasonic field during JECM drilling and milling of titanium alloys, demonstrating improvements in machining accuracy and process stability. Additionally, Ming et al. [29] experimentally examined the influence of jet orientation (horizontal, vertical, and oblique) on electrochemical machining, finding that horizontal jetting enhances depth uniformity and aspect ratio of microgrooves. Despite these advancements, significant challenges in JECM milling persist. A primary issue is the insufficient localization of the electrolyte jet field, which is largely responsible for the limited processing accuracy. The spreading, reflection, and free-flowing characteristics of the impinging jet (illustrated in Figure 1) hinder the achievement of highly localized material removal, often resulting in stray corrosion and overcutting. Furthermore, precise milling of holes and grooves with aspect ratios exceeding one remains difficult due to the inherent distribution of the energy field. Consequently, JECM is generally restricted to producing shallow microgrooves or micro-orifices with relatively low aspect ratios.

In addressing the aforementioned challenges, this study proposes a novel milling technique termed periodically thinning jet electrochemical milling (PT-JECM). This method entails the in situ periodic reduction in the jet diameter corresponding to the increasing depth of the machined grooves or holes. As a result, the width or diameter of the features exhibits minimal enlargement, or remains constant, during the fabrication of high aspect ratio structures. This contrasts with conventional JECM milling and drilling methods, which typically produce an increase in width or diameter exceeding half the depth. The present research further evaluates the viability of the proposed milling strategy through an integrated approach combining numerical simulations and experimental validation.

## 2. Processing Principles and Simulation

### 2.1. Concept of Periodically Thinning Jet Electrochemical Milling

The proposed methodology of periodically thinning the jet in electrochemical milling is delineated as follows: Throughout the milling operation, as the depth of machining escalates, the diameter of the jet beam is systematically diminished in a series of stages. Figure 2a depicts the conceptual framework of the periodically thinning jet electrochemical milling technique, specifically applied to the electrochemical milling of microgrooves.

(1) Stage 1: Use a jet with an initial beam diameter of d_1_ to perform electrochemical milling under the given process parameters, machining an initial groove with a depth of Z_1_ and a width of W_1_.

(2) Stage 2: Replace the nozzle in situ with a finer jet beam of diameter d_2_ and continue milling along the original tool path. During the process, after each scanning pass, the nozzle is fed downward by a value of ∆Z. When the groove depth Z_2_ reaches a certain value (at which point the groove width is W_2_), machining is paused and the process moves to the next stage. The groove width increment is kept within 10 μm to ensure that while the microgroove depth increases, the microgroove width does not expand, i.e., W_2_ is approximately equal to W_1_.

(3) Stage 3: Following a similar method to Stage 2, the jet beam diameter is progressively reduced in stages while continuing the milling process, until the designed groove depth Z_n_ (at which point the groove width is W_n_) is reached or the groove is fully penetrated. Figure 2b illustrates the tool path of the nozzle using the periodically thinning jet electrochemical milling method.

The subsequent analysis employs numerical simulation techniques to investigate the evolution of the flow and electric field associated with the aforementioned machining strategy.

### 2.2. Simulation Analysis

#### 2.2.1. Simulation Model Establishment and Solution

In order to elucidate the principle of stepwise contour removal within the context of the periodically thinning jet electrochemical milling technique, multiphysics simulation software, specifically COMSOL Multiphysics 6.1, is employed for analytical simulation. To streamline the simulation framework, a two-dimensional axisymmetric model representing the cross-sectional profile of the microgroove is developed. The subsequent assumptions are posited:(1)The electrolyte is considered a continuous, incompressible, and viscous fluid;(2)The flow of the electrolyte is axisymmetric;(3)The nozzle is rigid.

The two-phase flow (level set) module utilized in fluid dynamics, along with the primary current distribution, facilitates simulation within the software framework. The simulation process encompasses several key steps: the establishment of the physical model, the formulation of the mathematical model, mesh generation, equation resolution, and the extraction of the simulation results. Figure 3a illustrates a schematic representation of the physical model employed in the simulation, wherein domain I denotes the electrolyte, domain II signifies air, the nozzle functions as the cathode, and the initial groove is designated as the anode. A simplified version of the physical model is depicted in Figure 3b. In this model, boundary 1 corresponds to the electrolyte inlet, boundary 2 represents the initial interface between the electrolyte and air, boundaries 3 and 4 are identified as the electrolyte outlets, and boundary 5 is classified as the anode boundary. For the purposes of the simulation, the inner diameters of the cathode nozzle are specified as 350 μm and 500 μm. The boundaries and parameter settings of the model are shown in Table 1.

Since the microgroove model is simplified to a two-dimensional axisymmetric cross-sectional model, with a scanning speed of 100 μm/s, the machining time t is used to replace dynamic scanning machining, selecting a specific moment during the scanning process. 

The machining time t is equal to the ratio of the nozzle beam diameter d to the scanning speed of 100 μm/s, i.e.,t = dv^−1^
(1)

The steady-state current conservation equation is used to describe the electric field distribution:∇(σ∇φ) = 0(2)
where σ is the conductivity of the electrolyte, and φ is the electric potential. Combining the boundary conditions (the anode workpiece potential φ = V, the cathode nozzle grounded φ = 0), the current density distribution is calculated asJ = −(σ∇φ)(3)

In the simulation analysis, the geometric deformation of the material is determined by the “Deformed Geometry” module, which is used to model the time-dependent dissolution of the anode material. This model applies to boundary 5, where the material removal rate is based on Faraday’s law:Vn = ηM(z_a_ρF)^−1^Jn = ηV_sp_J_n_
(4)

In the equation, the surface normal material removal rate v_n_ depends on the current efficiency η, molar mass M, valence z_a_, density ρ, and the normal current density J_n_ = J × n; F is the Faraday constant.

In the DC machining simulation, the processing voltage is 25 V, the electrolyte inlet pressure is 1 MPa, and the maximum time step of the solver is set to 1 × 10^−2^ s. The simulation results are as follows.

#### 2.2.2. Simulation Results Analysis

Figure 4a,b depict the distribution of current density during microgroove machining utilizing nozzles with diameters of 350 µm and 500 µm, respectively. Notable disparities in current density distribution are observed between the bottom and sidewall regions of the microgroove. When employing the 350 µm nozzle, the current density is predominantly uniformly distributed across both the bottom and sidewalls of the microgroove, resulting in a lower overall current density. This uniformity contributes to a more gradual increase in groove width, thereby facilitating the attainment of a higher aspect ratio. Conversely, the use of the 500 µm nozzle leads to a concentration of current density along the sidewalls of the microgroove, which enhances the lateral material removal rate and further increases the groove width. These findings suggest that, when maintaining consistent machining parameters, a reduction in nozzle diameter promotes an enhancement in the aspect ratio of the microgroove.

Figure 5 depicts the distribution of microgroove depth along the arc length for two different nozzle diameters. In the case of the 350 µm nozzle machining condition, there is minimal lateral material removal observed. Prior simulations of current density corroborate that the machining area associated with the smaller nozzle is more focused, which results in a reduction in material removal from the sidewalls. This phenomenon enhances the microgroove depth while preserving a narrower width, thereby achieving a higher aspect ratio. Conversely, the use of a larger nozzle leads to the formation of a wider microgroove, which in turn diminishes the aspect ratio. This result provides an important theoretical basis for subsequent experiments.

## 3. Experiments and Methods

### 3.1. Electrochemical Measurement Setup

The electrochemical characteristics were measured in a three-electrode system using an electrochemical workstation (CHI604E, CH Instruments, Shanghai, China), as shown in Figure 6. A platinum plate was used as a counter electrode (CE). The reference electrode (RE) employed was Hg/Hg_2_Cl_2_, which was in contact with the electrolyte via a salt bridge to reduce the liquid junction potential. The Vit1 (10 mm (W) × 1 0 mm (L) × 2 mm (T)) (Zr41.2Ti13.8Cu12.5Ni10.0Be22.5) was insulated with epoxy resin as the working electrode (WE), and only 1 cm^2^ of surface area was exposed to the electrolyte. The open circuit potential (OCP) was monitored in solution until a stable surface state was reached before the electrochemical measurements.

This paper presents a comparative analysis of the anodic polarization behavior of surface stabilized Vit1 alloys when immersed in four different electrolytic media (NaNO_3_, NaCl, NaOH, and H_2_SO_4_). And the most suitable solution was electrochemically analyzed at different concentrations.

### 3.2. Experimental Setup and Materials

The experimental configuration utilized in this study is illustrated in Figure 7. It comprises an X-Y-Z motion platform. The workpiece under investigation is a block made of Vit1, a zirconium-based metallic glass, with dimensions of 20 mm in width, 20 mm in length, and 2 mm in thickness. This workpiece is affixed to the X-Y platform and serves as the anode, connected to the positive terminal of the electrolysis power supply. The nozzle, constructed from SUS 304, is positioned on the *Z*-axis of the motion platform, which boasts a displacement accuracy of ±0.1 µm. The coordination of positioning, displacement, adjustment of the anode/cathode gap, and trajectory control during the processing is managed through computer control of the *X*, *Y*, and *Z* axes of the motion platform. Additionally, the system is equipped with an electrolyte circulation and filtration mechanism, with the electrolyte pressure being maintained at a constant level via a high-pressure nitrogen tank. This setup is designed to ensure that the speed of the electrolyte jet remains as stable as possible throughout the processing phase. The machining process is powered by a programmable DC power supply (IT6122, Jiaozuo, China). The primary experimental conditions and parameters are detailed in Table 2.

In the context of gradual diameter reduction through jet electrochemical milling, the operation involves the utilization of multiple jet nozzles for clamping and switching purposes. To enhance the efficiency of nozzle switching and minimize installation inaccuracies, this research implements a mounting plate that secures three nozzles of varying outer diameters to the *Z*-axis of the motion platform. Throughout the milling process, each nozzle is linked to a common pressure tank, thereby ensuring that all nozzles are supplied with uniform input pressure.

### 3.3. Machining Effect Evaluation

In the tapered beam methodology utilized for microgroove machining experiments, the primary assessment metrics for the fabricated microgrooves encompass the aspect ratio (AR), the microgroove etching factor (EF), and the inclination of the microgroove sidewalls. The definitions of these parameters are illustrated in Figure 8. For instance, considering a nozzle with an inner diameter (d) of 500 μm, the formulas for calculating each evaluation parameter are provided below.AR = hw^−1^
(5)
where w is the microgroove width, h is the depth, and α represents the sidewall inclination of the microgroove.

The microgroove etching factor (EF) is the ratio of the groove depth to the groove overcut.EF = 2 h(w − 500)^−1^(6)

The microgroove sidewall inclination α isα = arctan[w(2 h)^−1^](7)

The three-dimensional morphology and profile of the microgroove were analyzed utilizing a laser confocal microscope (OLS5100, Olympus, Japan), which facilitated the measurement of its depth, width, and surface roughness (Ra). Additionally, the surface morphology of the microgroove was characterized through the use of a profilometer (SOBEK-AM600, Dongguan, China).

## 4. Results and Discussion

### 4.1. Electrochemical Characterization of Vit1

#### 4.1.1. Polarization Curve Testing

Figure 9a presents a comparative analysis of the anodic polarization behavior of the surface-stabilized Vit1 alloy when immersed in four distinct electrolytic media, namely, 10 wt% NaNO_3_, 10 wt% NaCl, 10 wt% NaOH, and 10 wt% H_2_SO_4_ solutions, all maintained at a temperature of 20 °C. The experimental findings indicate that the NaNO_3_ electrolyte system demonstrates superior passivation performance. Within the designated passivation potential range, the alloy surface maintains a stable passivated state, characterized by a current density consistently below 10^−4^ A·cm^−2^, which suggests the formation of a dense and highly protective passivation film. Upon exceeding a potential of 2.15 V, entering the over-passivation region, the passivation film experiences a controlled rupture, accompanied by the systematic dissolution of the material. Conversely, the passivation region observed in the H_2_SO_4_ system is significantly narrower, with the passivation current density escalating to the 10^−3^ A·cm^−2^ range, indicative of the inadequate structural stability of the passivation film. The polarization curves for the NaOH and NaCl systems do not display the characteristic passivation plateaus. The combined effects of Cl^−^ and OH^−^ ions considerably compromise the structural integrity of the passivation film, resulting in an exponential increase in anodic current below 0.2 V, which ultimately leads to the non-selective dissolution of the material.

Figure 9b illustrates the influence of varying concentrations of NaNO_3_ solution on the polarization behavior of the amorphous alloy. As the concentration of the solution is increased from 5 wt% to 20 wt%, the range of passivation potential remains constant; however, there is a progressive increase in the passivation current density. This trend suggests an enhanced capacity to disrupt the passivation film on the surface of the amorphous alloy. Among the tested concentrations, the 15 wt% NaNO_3_ solution demonstrates an optimal equilibrium between passivation and dissolution kinetics, effectively disrupting the passivation film while simultaneously mitigating stray corrosion. Consequently, it is identified as the most favorable process parameter.

#### 4.1.2. Electrochemical Impedance Spectroscopy (EIS) Analysis

Electrochemical impedance spectroscopy (EIS) tests were performed at three typical DC potentials (−0.1 V, 1 V, and 2.1 V) in NaNO_3_ electrolyte, and the fitted parameters are listed in Table 3.

At a potential of −0.1 V, the initial surface develops a monolayer passivation film, which typically demonstrates a relatively dense structure. At this potential, the charge transfer resistance (R_2_) reaches its maximum value, signifying that the passivation film possesses robust protective capabilities and high stability. As the polarization potential is increased to 2.1 V, there is a notable reduction in the impedance modulus, and the R_2_ decreases to 322.6 Ω·cm^2^, indicating the onset of material dissolution in the solution. The stability of the passivation film progresses through three distinct stages, namely, formation, stabilization, and breakdown, as the potential increases. The 15 wt% NaNO_3_ solution effectively mitigates stray corrosion of the amorphous alloy by balancing the dynamics of passivation and dissolution.

### 4.2. Effects of Machining Parameters

#### 4.2.1. Effect of Machining Voltage

The influence of external voltage is significant in determining machining precision, surface integrity, and morphological attributes throughout the electrochemical machining process. To examine the impact of voltage on microstructural morphology and geometric configurations, the experimental setup utilized specific parameters, including a supply pressure of 1 MPa, a machining gap of 200 μm, a nozzle traverse speed of 150 μm/s, and an applied voltage range of 15 to 30 V.

Figure 10a illustrates the morphology of microgrooves and their corresponding cross-sectional profiles under varying voltage conditions. At voltage levels of 15 V and 20 V, the microgroove profiles are relatively indistinct, exhibiting widths of 744 μm and 763 μm, and depths of merely 94.3 μm and 110 μm, respectively. An increase in voltage to 25 V enhances the anodic polarization effect, which facilitates the controlled rupture of the passivation film and significantly accelerates the material dissolution rate; as a result, the microgroove width and depth expand to 790 μm and 158 μm, respectively. Further elevating the voltage to 30 V leads to additional increases in microgroove dimensions, with widths and depths reaching 836 μm and 173 μm, respectively. This phenomenon can be attributed to the fact that at lower voltage levels, the passivation film on the surface of the sample does not undergo continuous dissolution, resulting in reduced material removal and shallower microgroove profiles. Figure 10b,c further demonstrate that as voltage increases, both the depth and width of the microgrooves progressively enlarge, concurrently enhancing the aspect ratio. However, a pronounced overcut phenomenon is observed between the voltage levels of 25 V and 30 V. Consequently, 25 V has been identified as the optimal applied voltage for jet electrochemical milling.

#### 4.2.2. Effect of Machining Gap

The machining gap is defined as the distance between the bottom of the cathode nozzle and the surface of the workpiece. This parameter significantly influences the intensity of the electric field during the machining process, necessitating a thorough examination of its impact on the results of jet electrochemical milling. In the present study, the following process parameters were employed: a supply pressure of 1 MPa, a scanning speed of 150 μm/s, a machining voltage of 25 V, and a machining gap ranging from 100 to 400 μm.

The relationship between the machining gap and the formation of microgrooves is illustrated in Figure 11. As the machining gap increases from 100 μm to 400 μm, there is a corresponding decrease in the depth, width, and aspect ratio of the microgrooves. This trend can be attributed to the increased gap, which results in a reduction in current density and a weakening of the electric field intensity, ultimately diminishing the efficiency of material removal. At a machining gap of 100 μm, the concentration of current density is excessively high, which can lead to short circuits and surface ablation, resulting in significant overcutting. In contrast, at a gap of 400 μm, the current density becomes too low, causing a marked reduction in the material removal rate. Notably, at a machining gap of 200 μm, the process demonstrates stability without the occurrence of short circuits, indicating that this parameter strikes an optimal balance between machining efficiency and the quality of the formed features.

#### 4.2.3. Effect of Nozzle Traverse Speed

In contrast to stationary micropores, the process of material removal in microgrooves is influenced by the relative motion between the tool electrode and the workpiece. An increase in the speed of the tool electrode can lead to substantial alterations in the distribution of the electric field, whereas a significantly reduced speed may lead to the buildup of reaction byproducts, which can adversely affect the quality of the machining process. This study employed the following process parameters: a supply pressure of 1 MPa, a machining voltage of 25 V, a machining gap of 200 μm, and a nozzle traverse speed ranging from 50 to 200 μm/s.

Figure 12a presents the microgroove morphology obtained at various nozzle traverse speeds of 50, 100, 150, and 200 μm/s. As illustrated in Figure 12b,c, the maximum values for microgroove depth, width, and aspect ratio are observed at a nozzle traverse speed of 50 μm/s. However, the prolonged dwell time of the nozzle on the workpiece surface exacerbates secondary stray corrosion, resulting in significant overcutting. At a traverse speed of 100 μm/s, the microgroove bottom surface exhibits a relatively smooth texture with well-defined boundary contours, achieving groove dimensions of 810 μm in width and 214 μm in depth, which indicates a superior quality of formation at this speed. As the nozzle traverse speed increases further, a gradual reduction in microgroove depth is noted, accompanied by a blurring of the edge contours. At a speed of 200 μm/s, the rapid movement induces considerable fluctuations in current density, which leads to uneven and inadequate anodic material removal. Consequently, a traverse speed of 100 μm/s is identified as the optimal scanning speed for jet electrochemical milling.

### 4.3. Tapered Beam Jet Electrochemical Milling of Microgrooves Experiment

#### 4.3.1. The Effect of the First Diameter Reduction

A nozzle featuring an initial jet diameter of 500 μm was employed for a singular scanning operation to establish a preliminary groove. This process yielded a microgroove characterized by a depth of 214 ± 1 μm, a width of 810 ± 4 μm, and an aspect ratio of 0.265. Building upon these results, a jet electrochemical milling experiment was conducted, wherein nozzles with progressively smaller diameters ranging from 450 to 300 μm were utilized to execute precision machining along the identical tool path, thereby enhancing the refinement of the preliminary groove.

Figure 13a illustrates the three-dimensional morphology and profile of microgrooves created using nozzles of varying diameters. The application of a 450 μm nozzle results in microgroove sidewalls that exhibit minimal step-like features; however, this configuration leads to an increase in groove width to 950 μm, which is indicative of excessive width expansion. In contrast, Figure 13b demonstrates that when utilizing 400 μm and 350 μm nozzles, the groove width remains relatively constant, while the depths attain values of 506 μm and 531 μm, respectively. This observation suggests that nozzle diameter is a critical parameter influencing material removal behavior. Conversely, the use of a 300 μm nozzle introduces pronounced step-like features on the microgroove sidewalls. This occurrence can be attributed to the non-uniform distribution of current density within the jetting region associated with smaller-diameter nozzles. Specifically, the current density is maximized directly beneath the nozzle, resulting in the highest rate of material removal. Consequently, this leads to excessive etching at the center of the groove bottom, thereby producing an uneven microstructure.

Figure 13c presents the etching factor and aspect ratio of microgrooves produced using nozzles of varying diameters. A decrease in nozzle diameter from 450 μm to 350 μm corresponds with a progressive increase in the etching factor, which reaches its peak at the 350 μm nozzle. Consequently, the 350 μm nozzle was identified as the optimal diameter for this experimental investigation. Following processing with the 350 μm nozzle, the resultant microgroove exhibited a width of 811 ± 4 μm, a depth of 531 ± 2 μm, an etching factor of 3.41, an aspect ratio of 0.655, and a sidewall inclination angle (α) of 0.65 radians.

#### 4.3.2. The Effect of the Second Diameter Reduction

In order to improve the etching factor and the aspect ratio while eliminating the stepped profile, a secondary diameter reduction jet electrochemical milling process was employed. Figure 14a illustrates the surface morphology and cross-sectional profile of microgrooves that were processed using various nozzle diameters during this second reduction phase. The width of all microgrooves consistently ranged from 810 μm to 820 μm, with no notable increase in groove width observed. The use of nozzles with diameters of 300 μm and 250 μm successfully eradicated the stepped traces on the sidewalls. This enhancement is largely attributed to the nozzle’s penetration into the microgroove, which augmented the current density along the sidewalls, thereby facilitating the removal of the stepped traces and resulting in smoother sidewalls. Conversely, the application of nozzles with diameters of 200 μm and 180 μm resulted in a pronounced disparity between the nozzle diameter and the initial groove width (810 μm), leading to the formation of a V-shaped structure within the microgroove.

Figure 14c illustrates the etching factor and aspect ratio of microgrooves produced using nozzles of varying diameters. The 300 μm nozzle demonstrated frequent sparking, which resulted in a reduced number of machining cycles and, consequently, a lower etching factor and aspect ratio. In contrast, the 180 μm and 200 μm nozzles experienced inadequate charge transfer within the electrolyte, leading to diminished current and subsequently lower material removal rates. This condition resulted in minimal groove depth and the lowest observed etching factors. Conversely, the 250 μm nozzle facilitated greater groove depth, achieving an etching factor of 4.23 and exhibiting a lower sidewall inclination without noticeable stepped traces. Therefore, in the subsequent diameter reduction phase of the jet electrochemical milling process, the 250 μm nozzle was selected. The resultant microgroove exhibited a width of 811 ± 3 μm, a depth of 656 ± 2 μm, an etching factor of 4.23, and an aspect ratio of 0.83. The sidewall inclination angle (α) was measured at 0.55 radians, with minimal stray corrosion observed along the edges and a high degree of machining consistency.

### 4.4. Comparative Analysis

Utilizing a nozzle diameter of 500 μm and maintaining consistent processing parameters, conventional jet electrochemical milling was performed with a machining gap of 200 μm and a single feed depth of 60 μm to achieve a groove depth that is comparable to that attained through the diameter reduction method. As illustrated in Figure 15a, the three-dimensional morphology of the microgroove produced by the conventional technique reveals that the groove depth reached 555 μm, while the width expanded to 1085.6 μm. In comparison to the conventional method, the periodically thinning jet electrochemical milling enhanced the etching factor from 1.896 to 4.318, reflecting an increase of 128%. Furthermore, the increment in groove width was minimized from 275 μm to 1 μm, effectively mitigating excessive widening. Additionally, the inclination of the sidewall was reduced from 0.77 rad to 0.54 rad, indicating a 30% improvement, while the aspect ratio increased from 0.51 to 0.83, signifying a 63% enhancement. The experimental results and simulation results show significant consistency in terms of principle and characteristic trends. The core principles of the new method revealed by the simulation were directly verified in the experiment.

Figure 15e depicts the morphological characteristics of the groove profile following jet electrochemical milling, employing both the diameter reduction technique and the traditional method. In the conventional approach, the width of the groove exhibits a continuous increase throughout the machining process. Conversely, the diameter reduction method preserves a relatively stable groove width, while the depth incrementally increases with each successive scan.

This indicates that the diameter reduction technique is effective in facilitating the machining of deep and narrow grooves, thereby substantially improving machining accuracy.

## 5. Conclusions

This study focuses on jet electrolytic milling by gradually reducing the diameter of the processing nozzle in order to obtain grooves with a large aspect ratio. The main conclusions are summarized as follows:The gradual reduction in the nozzle diameter effectively regulated the increase in groove width while facilitating a continuous enhancement in groove depth. This approach markedly minimized overcutting during the processing phase, leading to a more consistent improvement of the aspect ratio.Simulation analyses indicate that utilizing smaller nozzle diameters enhances the concentration of current density at the bottom of the microgroove, thereby minimizing the region of ineffective material removal. This phenomenon facilitates the accurate machining of deep and narrow grooves. Empirical findings corroborate the simulation results, suggesting that the method of progressively decreasing the nozzle diameter is particularly effective, offering a viable processing technique for the precise fabrication of microgrooves with a high aspect ratio.In comparison to conventional techniques, the etching factor of microgrooves produced through the gradual reduction in nozzle diameter method exhibited an increase from 1.896 to 4.318, representing a 128% enhancement. Additionally, the increment in groove width was significantly reduced from 275 μm to 1 μm, indicating an almost complete reduction of 100%. Furthermore, the sidewall tilt angle was decreased from 0.77 radians to 0.54 radians, thereby validating the efficacy of this novel process strategy.

In summary, this technology effectively addresses the technical bottlenecks encountered in traditional jet electrolytic machining when preparing complex microstructures in difficult-to-machine materials. It also deeply integrates the excellent material properties of amorphous alloys with the functional requirements of microchannels with large aspect ratios, providing critical processing technology support for the performance upgrade of core components in high-end equipment.

## Figures and Tables

**Figure 1 micromachines-16-00979-f001:**
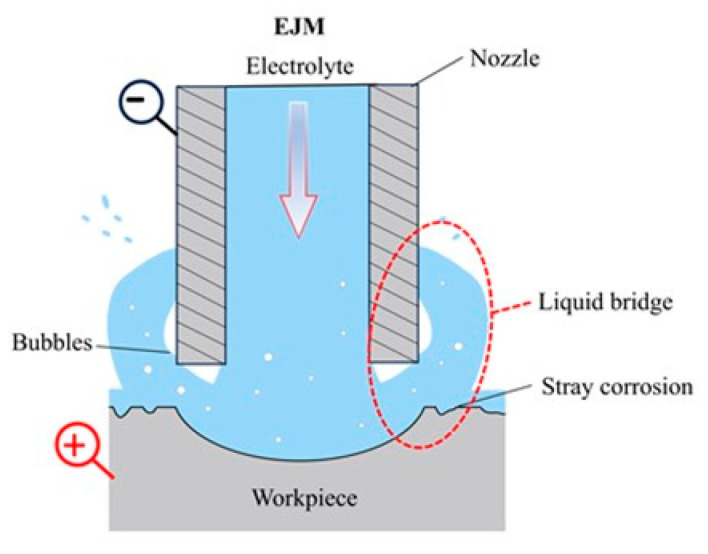
Schematic diagram of the conventional JECM.

**Figure 2 micromachines-16-00979-f002:**
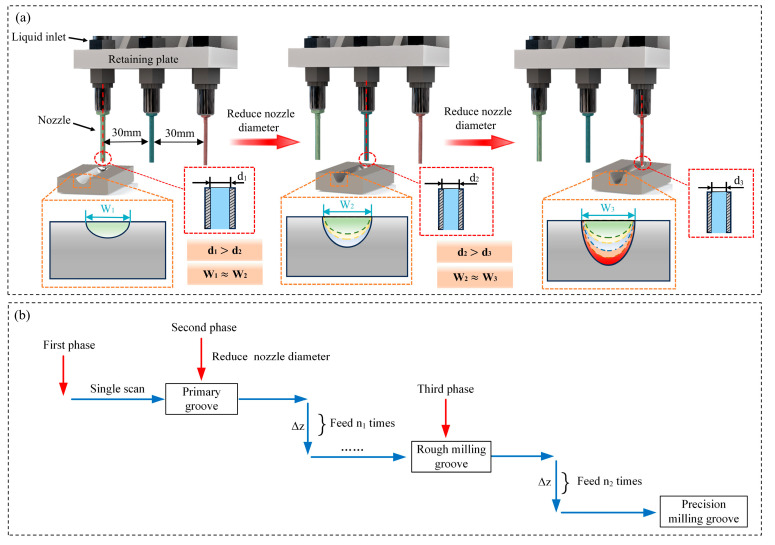
Principle of periodically thinning jet electrochemical milling for micro-slot fabrication: (**a**) machining strategy; (**b**) machining path planning.

**Figure 3 micromachines-16-00979-f003:**
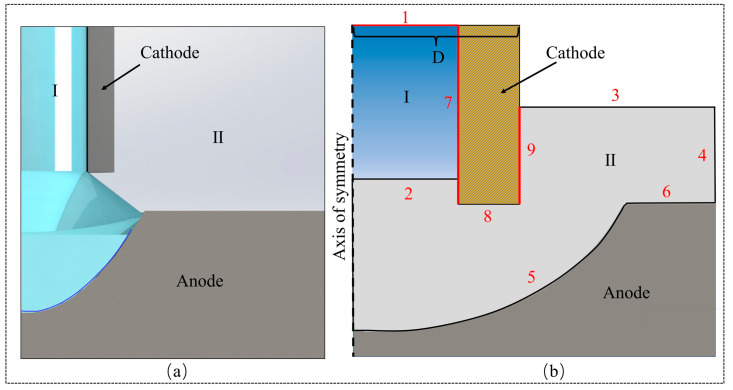
Schematic of the simulation model: (**a**) 3D model schematic; (**b**) physical model.

**Figure 4 micromachines-16-00979-f004:**
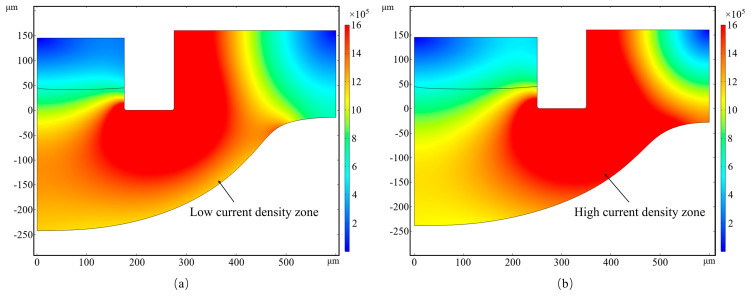
Current density simulation: (**a**) 350 μm diameter nozzle; (**b**) 500 μm diameter nozzle.

**Figure 5 micromachines-16-00979-f005:**
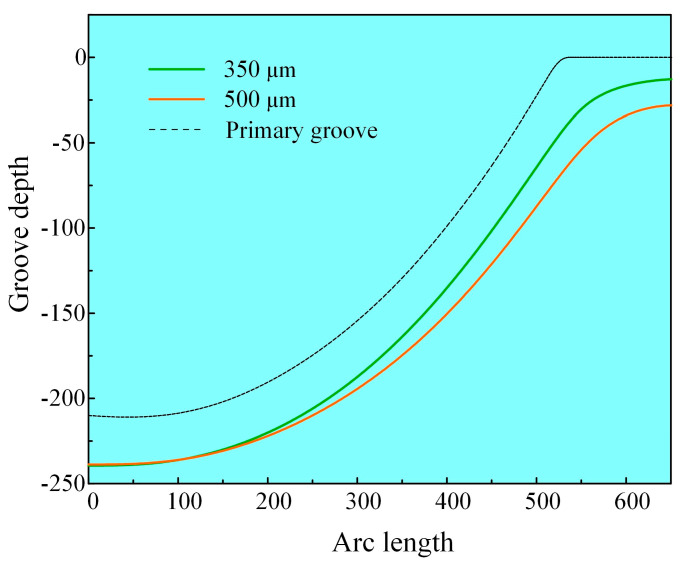
Simulation of machined microgroove profiles.

**Figure 6 micromachines-16-00979-f006:**
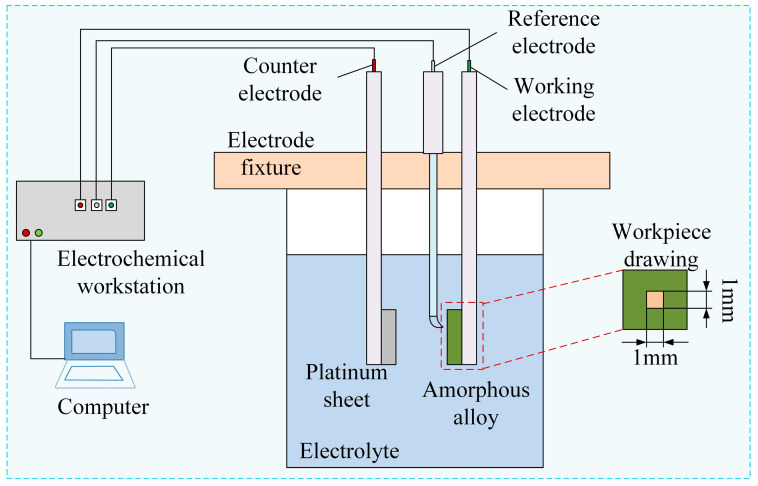
Electrochemical measurement setup.

**Figure 7 micromachines-16-00979-f007:**
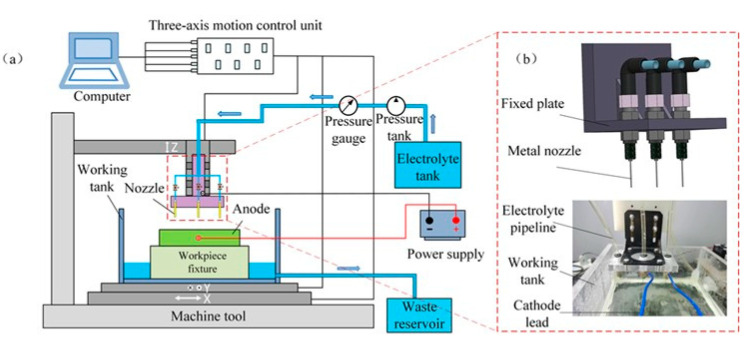
Schematic diagram of the tapered beam nozzle clamping and switching device: (**a**) machine tool structure diagram; (**b**) specific structure of cutting tools.

**Figure 8 micromachines-16-00979-f008:**
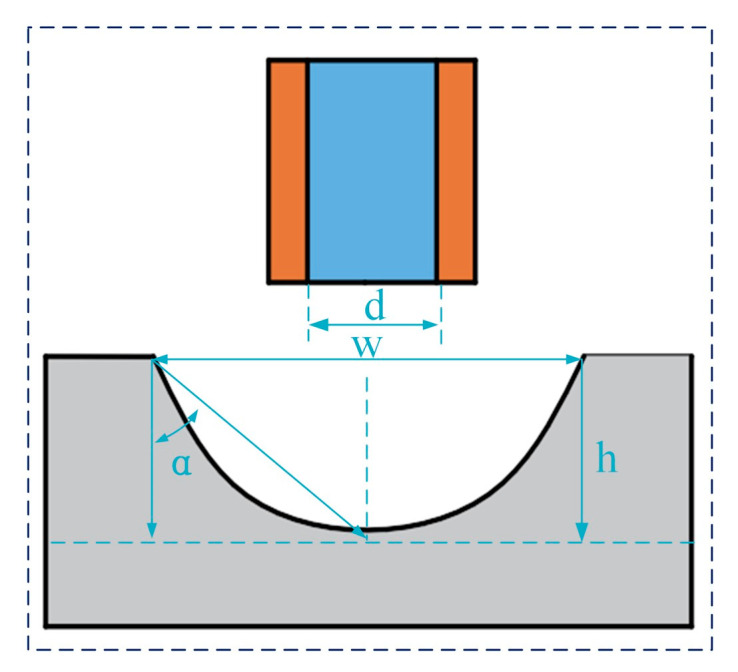
Machining result evaluation parameters.

**Figure 9 micromachines-16-00979-f009:**
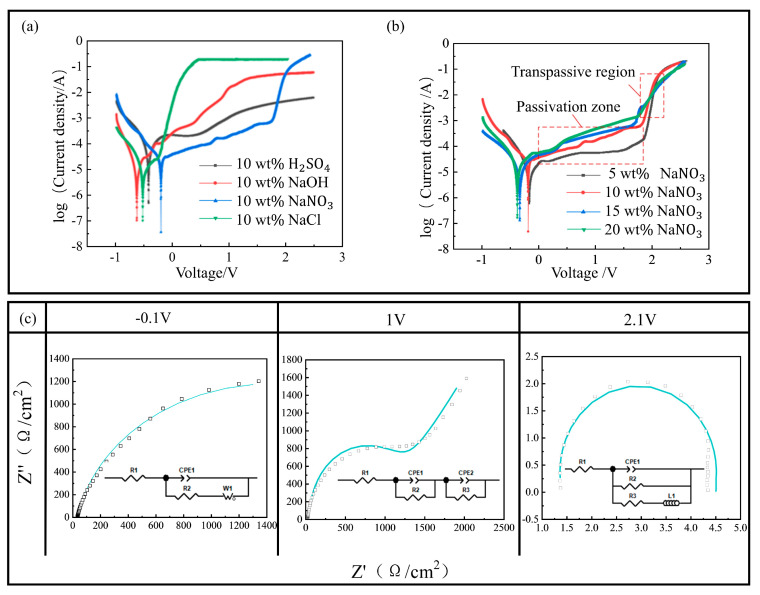
Polarization curve and electrochemical impedance spectroscopy (EIS) tests: (**a**) current efficiency under different solutions; (**b**) current efficiency of NaNO_3_ at different concentrations; (**c**) EIS results during polarization at different potentials.

**Figure 10 micromachines-16-00979-f010:**
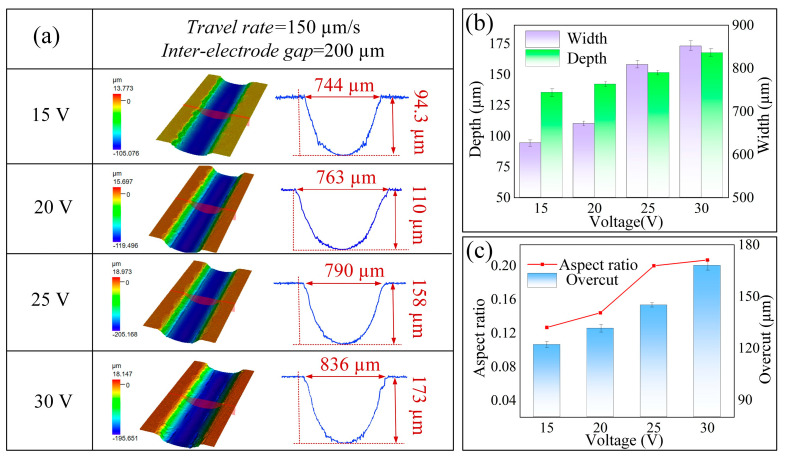
Effects of different voltages on machined microgrooves: (**a**) three-dimensional morphology and profile; (**b**) depth and width; (**c**) aspect ratio and overcut.

**Figure 11 micromachines-16-00979-f011:**
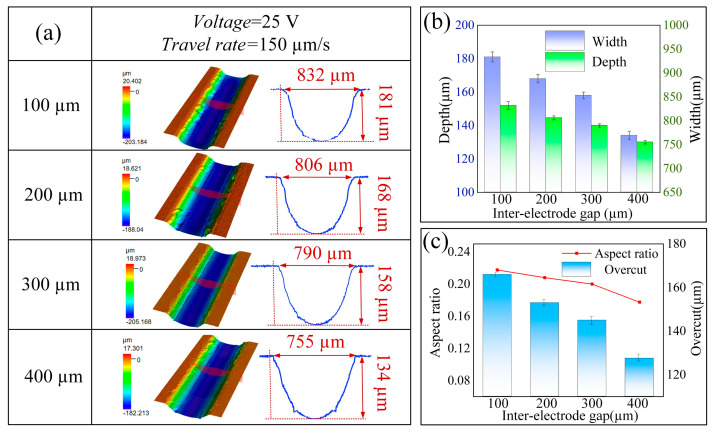
Effects of different machining gaps on machined microgrooves: (**a**) three-dimensional morphology and profile; (**b**) depth and width; (**c**) aspect ratio and overcut.

**Figure 12 micromachines-16-00979-f012:**
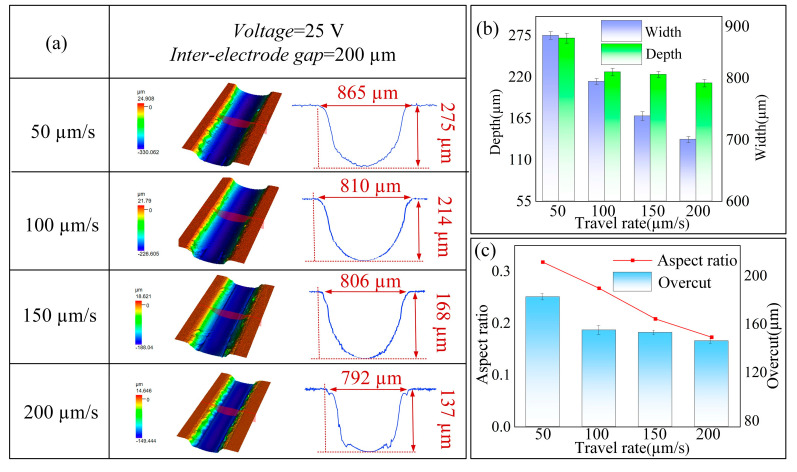
Effects of different feed speeds on machined microgrooves: (**a**) three-dimensional morphology and profile; (**b**) depth and width; (**c**) aspect ratio and overcut.

**Figure 13 micromachines-16-00979-f013:**
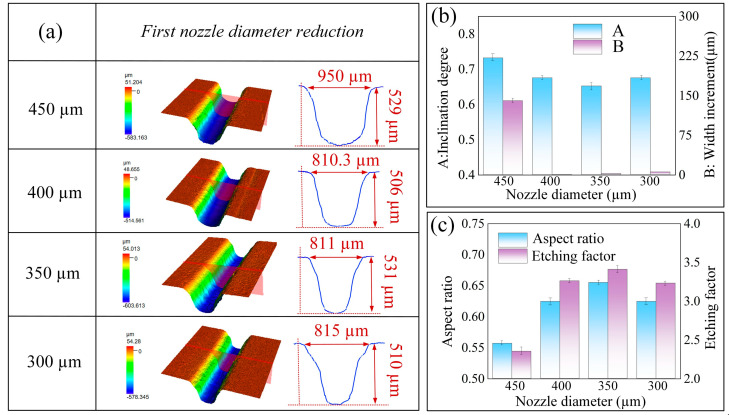
Effects of the first-stage tapered nozzle on machined microgrooves: (**a**) three-dimensional morphology and profile; (**b**) inclination and groove width increment; (**c**) aspect ratio and etching factor.

**Figure 14 micromachines-16-00979-f014:**
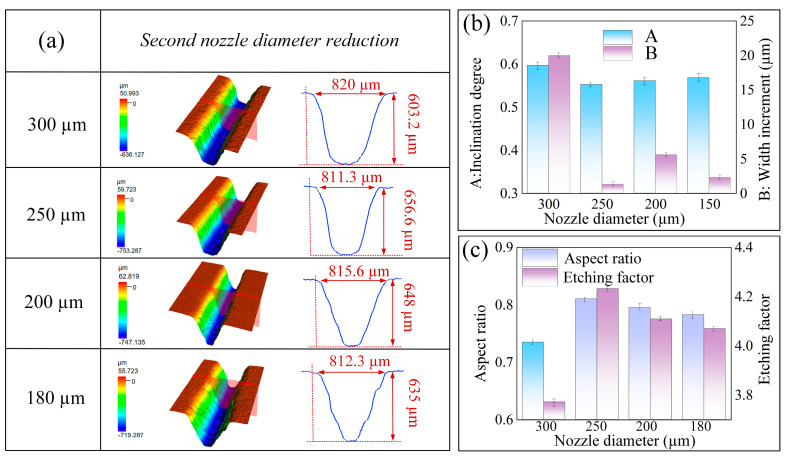
Effects of the second-stage tapered nozzle on machined microgrooves: (**a**) three-dimensional morphology and profile; (**b**) inclination and groove width increment; (**c**) aspect ratio and etching factor.

**Figure 15 micromachines-16-00979-f015:**
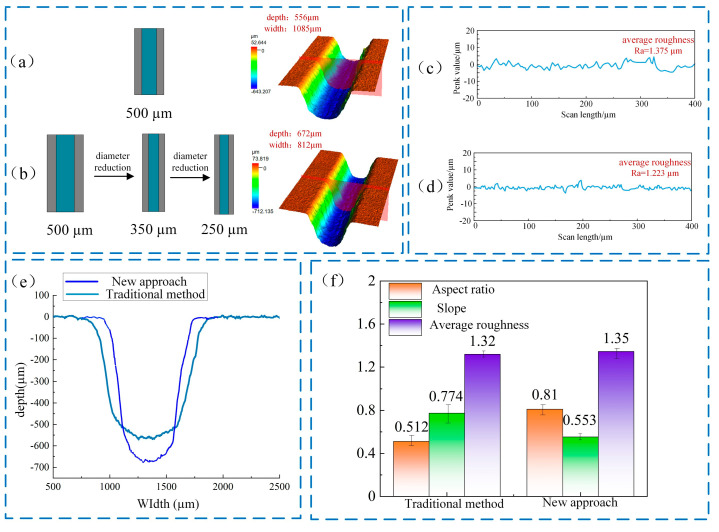
Comparison between traditional spray electrolysis and new methods: (**a**) microgroove morphology by traditional method; (**b**) microgroove morphology by decreasing jet diameter method; (**c**) traditional method of microgroove roughness; (**d**) reduced jet path microgroove roughness; (**e**) machined microgroove profiles using two methods; (**f**) aspect ratio, inclination, and roughness under different machining methods.

**Table 1 micromachines-16-00979-t001:** Boundary settings of model.

Physics	Boundaries	Conditions	
Geometry	D	175 μm	250 μm
Electric Field	7–9	Potential: φ = 0 Potential: φ = 25 V Inlet: velocity, U_0_ = 10 m/s Outlet pressure, p_0_ = 0 Pa	
	5–6		
Flow Field	1		
	3–4		

**Table 2 micromachines-16-00979-t002:** Processing conditions for JECM.

Parameters	Value
Material	Zr41.2Ti13.8Cu12.5Ni10.0Be22.5
Tool electrode	SUS 304 nozzle
Electrolyte pressure (MPa)	1
Machining gap (µm)	100, 200, 300, 400
Nozzle travel rate (µm/s)	50, 100, 150, 200
Machining voltage (V)	15, 20, 25, 30

**Table 3 micromachines-16-00979-t003:** Electrochemical impedance spectrum fitting parameters.

Potential (V)	R_1_ (Ω cm^−2^)	C_1_ (μF cm^−2^)	R_2_ (Ω cm^−2^)	C_2_ (μF cm^−2^)	R_3_ (Ω cm^−2^)	L_1_ (H cm^−2^)
−0.1	1.953	0.905	1445	-	-	-
1	2.344	1.1	1063	0.6109	1.0 × 10^20^	-
2.1	1.301	1.141	362	-	322.6	9.768

## Data Availability

Data are contained within the article.

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
