# Peer review of "Fabricating High Aspect Ratio Amorphous Alloys Microgrooves by Using Periodically Thinning Jet Electrochemical Milling Method"

_micromachines, 2025, doi:10.3390/mi16090979_

Round 1
Reviewer 1 Report
Comments and Suggestions for Authors
This manuscript focuses on the research of a jet electrochemical milling process, which has an interesting theme. However, there are still the following issues that need to be addressed.
- How to understand the "periodically" in the process proposed by the authors? This process reflects a gradual reduction in nozzle diameter. How to demonstrate the periodicity of this process?
- How does Figure 3a represent a schematic diagram of a 3D model? It shows a two-dimensional surface
- In section of 2.2.1, the details of simulation modeling need to be supplemented.
- The testing equipment used in sections 4.1.1 and 4.12 needs to be specified. In addition, these two parts were not mentioned in the section 3, including the effect of subsequent analysis of process parameters, which are not reflected in the processing parameters shown in Table 1. The experimental design should be detailed in Section 3.
- It is suggested to add error bars to the experimental data in Figures 9 to 14.
- When analyzing experimental results, it is necessary to combine the simulation results from the previous section to mutually verify and improve the closeness between simulation and experiment.
- The title of Section 5 should be changed to "Conclusion"
- In the conclusion, the positive significance of this process for engineering applications is suggested.
Author Response
Please kindly review the attached file.

Reviewer 2 Report
Comments and Suggestions for Authors
The manuscript presents a well-structured and pertinent study on jet electrochemical milling (JECM). The authors introduce an innovative approach, PT-JECM, supported by comprehensive results. The study convincingly demonstrates that the proposed method holds significant potential and merits further investigation. Although the manuscript aligns with the scope of micromachines, there are some improvements needed before acceptance, It’s important to note that although the PT-JECM has shown improved version however, there is a genuine question that how present work is novel ? there has been a lot of investigations already considered in the domain. For example, here are some links provided that has similar work -
https://doi.org/10.1016/j.ijmachtools.2022.103859
https://doi.org/10.1016/j.ijmachtools.2022.103931
https://doi.org/10.1016/j.cirp.2024.04.051
https://doi.org/10.1016/j.jmatprotec.2024.118571
Authors must provide the differences of the novelty of the proposed work for the further processing.
Comments on the Quality of English LanguageGood
Author Response
Please kindly review the attached file.
